# GDF15, an Emerging Player in Renal Physiology and Pathophysiology

**DOI:** 10.3390/ijms25115956

**Published:** 2024-05-29

**Authors:** Samia Lasaad, Gilles Crambert

**Affiliations:** 1Department of Pediatrics, Icahn School of Medicine at Mount Sinai, New York, NY 10029, USA; samia.lasaad@mssm.edu; 2Centre de Recherche des Cordeliers, Institut National de la Santé et de la Recherche Scientifique (INSERM), Sorbonne Université, Université Paris Cité, Laboratoire de Physiologie Rénale et Tubulopathies, F-75006 Paris, France; 3Unité Métabolisme et Physiologie Rénale, Centre National de la Recherche Scientifique (CNRS) EMR 8228, F-75006 Paris, France

**Keywords:** acid/base homeostasis, K^+^ balance, cell proliferation, GDF15

## Abstract

These last years, the growth factor GDF15 has emerged as a key element in many different biological processes. It has been established as being produced in response to many pathological states and is now referred to as a stress-related hormone. Regarding kidney functions, GDF15 has been involved in different pathologies such as chronic kidney disease, diabetic nephropathy, renal cancer, and so on. Interestingly, recent studies also revealed a role of GDF15 in the renal homeostatic mechanisms allowing to maintain constant, as far as possible, the plasma parameters such as pH and K^+^ values. In this review, we recapitulate the role of GDF15 in physiological and pathological context by focusing our interest on its renal effect.

## 1. Background

Growth Differentiation Factor 15 (GDF15) is a growth factor and cytokine secreted in response to cellular stress conditions, belonging to the TGFβ family, discovered in 1997. It is also referred to as Macrophage Inhibitory Cytokine 1 (MIC-1) due to initial studies demonstrating its ability to inhibit lipopolysaccharide-activated macrophages [1]. Subsequent studies have highlighted various functions of GDF15, leading to alternative designations such as NSAID-activated gene 1 (NAG-1), Placental Transforming Growth Factor β (PTGFB), and Prostate-Derived Factor (PDF) [2,3,4]. In humans, the gene encoding GDF15 is located on chromosome 19p12–13.1 and spans 2746 base pairs. While murine and human genes exhibit only 70% homology, they both consist of two exons separated by an intron [5] (Figure 1A). GDF15 is synthesized as an inactive pro-hormone consisting of 167 amino acids with a molecular weight of 40 kDa. It is then dimerized in the endoplasmic reticulum before being activated by cleavage at a furin-like site, resulting in the formation of a 25 kDa peptide dimer (Figure 1B). The uncleaved form of GDF15 exhibits a high affinity for the extracellular matrix, where it forms a reservoir allowing rapid secretion into circulation upon proteolysis. Only the mature form is secreted [6]. GDF15 is highly expressed in the prostate and placenta, but it is also weakly expressed in various tissues such as the kidneys, lungs, liver, intestines, colon, pancreas, and mammary gland [5,7,8]. It has also been demonstrated that GDF15 plasma concentration is increased during gestation in mice [9]. In pregnant women, the increase in GDF15 is correlated to nausea and vomiting [10]. More generally, GDF15 expression is induced in response to cellular stress, which explains why it is frequently associated with many diseases including cancer, obesity, cachexia, cardiovascular events, kidney diseases, etc. (for review see [8]).

### 1.1. Signaling Pathways Related to GDF15

Although the signaling pathway induced by GDF15 is not fully elucidated, studies have shown that GDF15 is the ligand for the α-like receptor of the Glial-Derived Neurotrophic Factor (GDNF) family, GFRAL. The GFRAL receptor is a 44 kDa protein with a single transmembrane domain and a short cytoplasmic domain. The gene encoding the protein consists of six exons, located on chromosome 6p12.1 in humans, and the sequence is highly conserved in mice. This gene also codes for a truncated form of GFRAL, 27 kDa in size, possessing the GDF15 ligand-binding domains but lacking a transmembrane domain. This soluble form of GFRAL, whose function is not clear, may block the action of GDF15 by acting as a soluble receptor (for review see [11]). GFRAL expression is limited to the brain (area postrema and nucleus of the solitary tract) and is not found in peripheral tissues [12], although a recent study suggests that it could be more widely expressed than established before [13]. This point, however, needs to be better characterized since, for instance in kidneys, the mRNA of GFRAL is not found [12] and the reported immunolocalization mainly shows intracellular labeling. Upon binding to the receptor, the GDF15-GFRAL complex binds and phosphorylates the co-receptor tyrosine kinase RET. RET phosphorylation activates the AKT, ERK1/2, and PKC signaling pathways. It was proposed that GDF15 may also activate the SMAD signaling pathway, but it turns out that these observations were probably due to contamination of “purified” GDF15 by TGFβ. (for review see [14,15,16]).

The limitation of GFRAL receptor expression, despite the pleiotropic action of GDF15, suggests the presence of other receptors. In recent studies [17,18], it has been proposed that the tyrosine-protein kinase receptor ErbB2 is also a receptor for GDF15. ErbB2, also known as HER2 or NEU, belongs to the family of transmembrane tyrosine kinase receptors ERBB. This family comprises four receptors: HER1, HER2 (corresponding to the ErbB2 receptor), HER3, and HER4. The receptors exist as monomers or dimers. ErbB2 is a 185 kDa transmembrane receptor, consisting of 1255 amino acids, with a ligand-binding extracellular domain of 632 amino acids and an intracellular domain with tyrosine kinase activity of 580 amino acids [19]. In a physiological context, the ErbB2 receptor plays a significant role in cell proliferation and differentiation. However, its overexpression is associated with the development of several cancers. It has been demonstrated that in human breast and gastric cancer cells, GDF15 promotes tumor cell proliferation via activation of the ErbB2 receptor and the AKT and ERK-1/2 signaling pathways [17]. This association between the ErbB2 receptor and GDF15 has also been studied in the context of cervical cancer cells, demonstrating that GDF15 also activates ErbB2 in this context and the PI3K/AKT and MAPK/ERK signaling pathways [18]. Finally, it has been demonstrated in the literature that GDF15 and ErbB2 have been co-immunoprecipitated, however, further investigations are needed to establish whether there is a direct interaction between GDF15 and the ErbB2 receptor [18].

### 1.2. Regulation of GDF15 Expression and Plasma Concentration

Key intracellular proteins governing the transcription of the GDF15 gene include p53 and EGR-1 [20,21]. P53 is a tumor suppressor gene. It acts as a transcription factor, binding specifically to regulatory regions of genes. Under physiological conditions, p53 is expressed at low levels in the nucleus. During cellular stress, such as DNA damage, p53 can arrest cell cycle progression to allow DNA repair or induce apoptosis [22]. EGR-1 is also a crucial nuclear transcription factor involved in essential cellular processes including cell proliferation, differentiation, and apoptosis [23]. Recently, transcription factors from stress-induced signaling pathways, such as CHOP and ATF4, have also been identified as activators of GDF15 expression [24,25,26]. Therefore, it is not surprising that the plasma concentration of GDF15 increases during physiological stress and in pathological conditions. In humans, plasma concentration of GDF15 increases with factors such as age, smoking, drug consumption, and in response to metabolic stress like diabetes [26]. It has also been demonstrated that GDF15 increases in response to physical exertion, where it is produced by muscles [27]. Plasma levels of GDF15 also increase in most types of cancer, chronic inflammatory diseases, cardiovascular and renal diseases, severe infections, and can predict all-cause mortality [28,29,30,31]. Furthermore, studies have shown that plasma concentration of GDF15 increases with tissue injuries and the use of certain treatments, including metformin, as well as during anticancer therapies such as chemotherapy and ionizing radiation [32,33,34]. These data indicate that GDF15 is potentially an important biomarker in various pathologies. Nonetheless, depending on the studies and pathophysiological contexts, the elevation of plasma GDF15 concentration indicates either a favorable or an unfavorable prognosis. This underscores the need for further investigations on the mechanisms involving GDF15. Regarding kidney pathophysiology, GDF15 has been shown to be involved in different aspects that concern fibrosis, tubular integrity, cell proliferation, etc. (see below).

### 1.3. GDF15 and Kidney

The kidney consists of nephrons that are composed of a filter, the glomerulus, and a tubular system. If the glomerulus retains the blood cells and the macromolecules, it permits the passage of small molecules (low molecular weight proteins and peptides, glucides, water …) and ions. The role of the tubular system is to reabsorb these different components through complex mechanisms. For this purpose, the tubular system is segmented and it is possible to isolate these different parts, based on their morphology, in order to study their function. Figure 2A is an example of how these segments look like with the proximal convoluted tubules (PCT), the proximal straight tubules (PST), the thick ascending limbs of the Henle loops (medullary or cortical TAL), the distal convoluted tubules (DCT), the connecting tubules (CNT, and the collecting ducts (outer medullary or cortical CD). Under physiological conditions, GDF15 mRNA is expressed by the kidney itself, although at relatively low levels, all along the tubular segments of the nephrons (Figure 2B). The use of a GFP reporter mouse strain expressing this fluorescent protein under the GDF15 promoter confirmed its expression in different tubular segments and particularly in PST and in medullary CD [35]. Investigation of GDF15KO mice under normal conditions showed that it regulates both the production of renal extracellular matrix and promotes the proliferation of tubular epithelial cells via the MAPK signaling pathway [36]. However, these GDF15KO mice do not exhibit any electrolytic perturbation under normal conditions [37,38].

### 1.4. GDF15 in Diabetic Nephropathy

Diabetic nephropathy is a consequence of diabetes and at the origin of chronic kidney disease (CKD). Analysis of the single cell RNAseq databank (https://humphreyslab.com/SingleCell (accessed on 1 April 2024)) of patients suffering diabetic nephropathy [39] indicates that the expression of GDF15 is upregulated, mainly in principal cells of the collecting ducts. In murine models of type 1 and type 2 diabetes, GDF15KO mice exhibit more tubular lesions, interstitial damage, and more significant signs of glycosuria and polyuria compared to WT diabetic mice, suggesting that GDF15 is necessary for maintaining tubular integrity [40]. GDF15 levels are elevated in diabetic conditions, and Zhang X et al. revealed its protective effect against inflammation-induced podocyte injury [41]. Mechanistically, GDF15 inhibits the nuclear translocation of nuclear factor kappa B (NF-κB) through interaction with the ubiquitin ligase NEDD4L, thereby reducing inflammation and podocyte injury. Interestingly, urinary GDF15 levels predict renal function decline in diabetic kidney disease patients, independently of plasma GDF15 levels [42]. Interestingly, the opposite experiments using transgenic mice overexpressing GDF15 and treated with high fat diet and streptozotocin to induce diabetes were “protected” against the renal damages [43]. They exhibited reduced signs of renal dysfunctions (blood urea, serum creatinine, and histological damages) and less polyuria/polydypsia. In this study, GDF15 overexpression was shown to limit the inflammation induced by the AGE/RAGE pathway which is upstream of the activation of NF-κB. The elevation of GDF15 levels in diabetic animal models or patients is further heightened with metformin treatment [34,44] which may contribute to the metformin effects on lowering appetite and weight loss [45]. Stimulation of GDF15 expression by hepatocytes in response to metformin treatment involves endoplasmic reticulum stress and ATF4 and CHOP activation [45] and is correlated with metformin doses. This overproduction of GDF15 in response to metformin could participate in the decrease of appetite and food consumption in diabetic patients by activation of GFRAL which may lead to weight loss and ameliorate their glucose balance.

### 1.5. GDF15 in Chronic Kidney Disease (CKD) and Acute Kidney Disease (AKI)

In CKD, the urinary GDF15 level is increased, suggesting potential utility as a biomarker [46]. It was indeed shown that serum GDF15 levels in patients with renal diseases are elevated and are associated with comorbidities and pathological features [47,48]. Importantly, higher serum GDF15 levels predict adverse renal outcomes, including renal function decline and renal replacement therapy [47]. GDF15 deficiency exacerbates renal fibrosis, whereas GDF15 overexpression attenuates fibrotic changes, suggesting a protective role for GDF15 against renal fibrosis [49]. Mechanistically, GDF15 inhibits renal fibrosis by suppressing the transforming growth factor-beta 1 (TGF-β1)-mediated epithelial-mesenchymal transition (EMT) and fibroblast activation.

GDF15 seems also to be involved in the pathological processes following acute kidney injury (AKI), since its expression is triggered early in PST S3 and principal cells (PC) of CD [35]. Analysis of the available scRNAseq databank confirmed that GDF15 expression is strongly induced by ischemia/reperfusion and is related to the duration of both the ischemia and the reperfusion (IRI) periods (from [50], Figure 3A,B). Valiño-Rivas et al. [51] showed, in both models of IRI (by unilateral ureteral obstruction) and of CKD (treatment with folic acid), that GDF15 administration attenuates the renal effect of IRI by stimulating expression of Klotho, a nephroprotective factor (for review see [49]), reducing oxidative stress, inflammation, and apoptosis, thereby preserving renal function. Mechanistically, GDF15 activates the PI3K/Akt pathway and inhibits the NF-κB signaling pathway, leading to the suppression of pro-inflammatory cytokines and apoptotic mediators. Interestingly, kidney injury in the context of Cockayne syndrome, a rare degenerative disorder, is associated with GDF15 production by a specific subpopulation of renal proximal tubule cells [52]. This specific production of GDF15 may contribute to the cachexia observed in this syndrome. All together, these different studies indicate that GDF15 is elevated during acute and chronic kidney disease and protects the kidney, mainly, by limiting the inflammation.

### 1.6. GDF15 in Renal Cancer

GDF15 is a well-established hallmark of solid cancer tumors of various origins and can be targeted for cancer therapy [53]. Notably, a 2020 study by Jiang et al. [54] demonstrated that GDF15 is also induced by cancer treatments such as cisplatin, a drug known for its strong nephrotoxicity. Furthermore, recent research has shown that candesartan treatment, an angiotensin II antagonist, effectively protects against cisplatin-induced nephrotoxicity. Candesartan treatment is suggested to elevate serum GDF15 levels, leading to a reduction in markers of inflammation and oxidative stress, as well as protection against kidney injury, according to Güner et al. [55]. This effect is not observed when cisplatin is administered alone. GDF15 expression is significantly decreased in clear renal cell carcinoma (ccRCC) tumors, contrary to papillary renal cell carcinoma and chromophobe renal cell carcinoma [56]. In ccRCC, survival of patients is inversely correlated to GDF15 expression and in vitro experiments revealed that the presence of GDF15 limits the migration and invasion of the cells by activating ferroptosis.

## 2. GDF15 and Electrolyte Balance

Since the 19th century and the definition of the “milieu intérieur” and the establishment of the concept of homeostasis developed by Claude Bernard and, later, by Walter Bradfor Canon (for review see [57]), it is well understood that the parameters of the internal medium must remain stable to support normal cell life. Thus, complex mechanisms are involved to help maintain plasma parameters, like pH and ion concentrations, in their normal range despite the variable intakes that may occur throughout the day. The kidney is one of the main actors in this homeostatic process since its essential function is to balance the output (in the urine) with the input. As mentioned above, the function of the tubular system is to reabsorb or secrete small molecules or ions in a regulated way to equilibrate the different balances. Tubulopathies are pathologies that affect some parts of the tubules and disturb the processes of water or ion excretion, resulting in systemic diseases [58]. As we will see below, GDF15 has been described as participating in the establishment of acid/base and potassium balances by responding to ionic challenges such as acid load or K^+^ restriction. This growth factor therefore contributes to the equilibrium of the electrolytic balance by acting on the kidney.

### 2.1. Involvement of GDF15 in Response to Acidosis

The blood pH must be restricted to a very narrow range, from 7.36 to 7.44, whatever the dietary intakes. The modern Western diet is characterized by consumption of processed food with high intakes of proteins, high-fat dairy products, and high-sugar drinks correlating with development of metabolic disorders (diabetes, obesity). This type of diet has also strong but often underappreciated consequences on electrolyte balances. Indeed, modern humans eating a typical Western diet produce approximatively 50 mmol of acid/day whereas our ancestors were considered to be net base producers [59]. Adaptation to acid load is a complex physiological process that involves renal and extra-renal mechanisms. In the kidney, both the proximal and distal parts of the nephron participate in the transport of H^+^, NH_4_^+^, and HCO_3_^−^, contributing to the excretion of an acid load (for review see [60]). In the collecting duct, type A intercalated cells (AIC) play a major role in this process since they are equipped with an apical vacuolar H^+^-ATPase and a basolateral chloride/bicarbonate exchanger permitting proton secretion and bicarbonate reabsorption. The number of AIC was shown to increase after an acid load [61,62,63], which involved proliferation and/or transdifferentiation processes. Indeed, after being established that the number of AIC increased in response to acidosis, the question regarding the mechanisms involved in this response was investigated. The group of Dr Al-Awqati introduced the concept of cell plasticity by showing in a rabbit cortical collecting duct that type B intercalated cells (BIC that are responsible for bicarbonate secretion and proton reabsorption) could “convert” to AIC [61]. This mechanism of adaptation to acid load would have a clear advantage by decreasing the number of cells secreting bicarbonate and increasing the number of cells reabsorbing bicarbonate. The same group later identified a protein, the hensin [64], an extracellular matrix protein whose expression and binding to its receptor depends on the pH [65]. Indeed, under acidosis, hensin polymerizes in the extracellular matrix and activates the β1 integrin [66]. The deletion of hensin or of the β1 integrin [65] impedes the BIC conversion to AIC and leads to profound acidosis. However, the existence of such a mechanism cannot explain the increase in number of AICs under acidosis in the medullary collecting duct since BIC are not present in the medulla. In 2006, GDF15 was identified as one of the most upregulated genes in the medullary collecting duct in response to an acid load [67]. It was shown later that the stimulation of GDF15 expression in the collecting duct follows a specific kinetic, reaching a peak 3–4 days after the acidic challenge [68] before returning to baseline with prolonged exposure to the acid load. In this acidotic context, vasopressin (AVP) triggers GDF15 production in the principal cells of the collecting duct through a pathway involving AMPK isoforms, the stimulation of the Na,K-ATPase activity (that contributes to a decrease of the ATP/AMP ratio and stimulate AMPK), and p53 [38] (Figure 4A). GDF15 may then activate the ErbB2 receptor on the AIC which will trigger the cyclin D1-dependent cell division. Of note, the activation of the ErbB2 receptor may require its sensitization by calcium–calmodulin [69] after stimulation of the AIC vasopressin receptor, the V1aR, known to participate in the response to acidosis [70,71]. Interestingly, in the absence of GDF15, the response to an acid load is altered [68]. During the first day of the challenge, mice exhibited impaired adaptation and developed severe acidosis, which is correlated with an absence of AIC proliferation dependent on cyclin D1 expression. However, after two weeks of acid load, GDF15KO mice showed a recovery and exhibited similar physiological parameters (blood pH and bicarbonate) as the WT mice, indicating that another, GDF15-independent, process has been involved. As it revealed that this GDF15-independent process needs activation of Cyclin D3 and is correlated with a transverse proliferation instead of a longitudinal proliferation. GDF15 produced by the principal cells then activates the ErbB2 receptor and triggers the proliferation of AIC [38].

### 2.2. Involvement of GDF15 in Response to K^+^ Restriction

The kidney is the main actor in the external K^+^ balance [72]. It plays an essential role in maintaining plasma K^+^ level (for review see [73]) ensuring that urinary K^+^ excretion is equal to its daily intake. The consumption of salt and potassium have been completely modified these last years, passing from a rich-K/low NaCl diet in the hunter-gatherer population to the opposite in the modern, westernized population [74].

Renal K^+^ excretion occurs following glomerular filtration and through K^+^ transport along the renal tubule where it is the sum of secretion and reabsorption processes. Among these different transport processes in all the tubular segments, the ones that are regulated in the distal part are the most important to achieve the K^+^ balance. The collecting duct is capable of both reabsorbing and secreting K^+^. In this zone of the nephron, both K^+^ reabsorption and secretion are modulated by regulatory factors and hormones. An intake rich in K^+^ leads to an increase in its secretion and an inhibition of its reabsorption. Conversely, a low K^+^ intake induces increased reabsorption and inhibition of K^+^ secretion. In the collecting duct, K^+^ secretion and reabsorption are mediated by principal cells and type A intercalated cells (AIC).

In the context of potassium (K^+^) secretion within renal physiology, the establishment of a favorable transepithelial potential, primarily orchestrated through the activation of the epithelial Na^+^ Channel (ENaC) [75] is the primum movens of the K^+^ movement outside the cells. This activation facilitates the luminal efflux of K^+^ from principal cells via potassium channels, particularly the Kir1.1 channels, also known as ROMK. Notably, alongside ROMK, another potassium channel with a large conductance, the big K (BK) or Maxi K channel, significantly contributes to K^+^ secretion [76] in a flux- and Ca^2+^-dependent manner [77].

As for mechanisms promoting K^+^ reabsorption in response to K^+^ restriction, the main processes that have been described are 1/the stimulation of the expression and activity of the H,K-ATPase type 2 (HKA2) at the apical membrane of AIC [78] and 2/the increase in the number of type A intercalated cells to increase the surface area for K^+^ reabsorption [79,80,81,82]. The first process has been shown to involve, on one hand, reactive oxygen species and activation of Nrf2 transcription factor [83] and, on the other hand, the adrenal production of progesterone [84]. Regarding the increased number of AIC that express the HKA2 [85,86], it involves cell proliferation [80] and/or cell transdifferentiation [79,87]. GDF15 was suspected to participate in AIC proliferation in response to K^+^ restriction since it is one of the most upregulated genes in a mouse’s collecting duct under such condition [80]. We recently confirmed this observation and showed that GDF15 was involved in the proliferation process of AIC in response to K^+^ restriction [37]. We found that GDF15 expression increased in all renal tubular segments, as well as in parts of the intestine and colon of mice in response to a low-K^+^ diet leading to a systemic elevation of GDF15 concentrations in both plasma and urine. The trigger of this overexpression of GDF15 in response to K^+^ depletion is not yet known. However, it is established that this condition induced an oxidative stress with production of reactive oxygen species [88], at least, in renal tubular cells like the principal cells of the collecting duct, which may suggest a dysfunction of the mitochondria. Of note, mitochondria dysfunction has been involved in the stimulation of GDF15 expression [89]. In GDF15KO mice, we observed a delay in the renal adaptation to a low K^+^ diet compared to WT. This delay is related to the absence of AIC proliferation in GDF15KO mice whereas WT mice increased their number of AIC by 15%. The action of GDF15 on AIC is sensitive to mubritinib, a specific inhibitor of the ErbB2 receptor. Interestingly, we also observed that the expected overexpression of HKA2 in response to a low K^+^ diet was completely blunted in GDF15KO mice, suggesting that GDF15 also participate in the stimulation of this pump. Moreover, we observed that the HKA2 is also important for cell proliferation, as the HKA2KO mice did not increase their AIC number under a low K^+^ diet as the WT did. A possible explanation for the role of HKA2 in the proliferation process of AIC is the fact that these cells have a very low expression of Na,K-ATPase [90]. During cell division, the increased rate of the cell metabolism, in order to double the DNA, protein, and lipid contents, dramatically increases the production of H^+^ that have to be extruded. In most of the cells, this is done by a Na/H-exchanger [91] that requires a Na,K-ATPase to generate and maintain Na^+^ gradient. Moreover, cell division also implies an increase in the uptake of K^+^ [92,93] in order to maintain the cell membrane potential, here again, this is normally performed by the Na,K-ATPase. In AIC, the presence of the HKA2 that has the possibility to extrude H^+^ from the cells and to take up K^+^ into the cell could replace the Na,K-ATPase-Na/H-exchanger system. The role of HKA2 in the proliferative process has already been demonstrated in cancer cells [94]. During acidosis, the role of vasopressin was established in both the production of GDF15 by the principal cells and in the proliferation process of the AIC (see above). Here, in the context of K^+^ depletion, such a link has not been investigated, but the vasopressin response is reported to not be different [95] or even blunted during K^+^ depletion [96]. It is therefore unlikely that vasopressin was involved in AIC proliferation in this context on the contrary to acidosis. K^+^ depletion has a strong effect on muscle cells since it is one of the most important reservoirs of K^+^ in the body that could be used to store or release K^+^ in an insulin-dependent manner [97]. Interestingly, the impairment of renal K^+^ handling as observed in GDF15KO mice decreased the size of the muscle fibers, suggesting that the mass of the muscle is reduced. This is confirmed with the observation that markers of muscle degradation such as TRIM63 and Fxbo32 are overexpressed in muscle of GDF15KO mice under K^+^ restriction. Therefore, GDF15 triggers the proliferation of AIC by interacting to ErbB2 and stimulates the expression of the HKA2 that permit the AIC division to come to an end. These data suggest that there is a pathway from GDF15 to cell proliferation that involves HKA2 (Figure 4B).

A possible effect of GDF15 on the K^+^ balance concerns pregnancy, a physiological condition of renal K^+^ retention involving the HKA2 [98]. Indeed, it is well-established that plasma GDF15 is elevated in humans [10] as in mice [9] during the gestation and one may draw the hypothesis that it contributes to increase the number of AIC and therefore help at retaining K^+^. This hypothesis however remains to be proven.

## 3. Conclusions/Perspective

GDF15 is an emerging factor that displays pleiotropic functions involved in many different pathological states (metabolic syndrome or cancer, for instance) and physiological processes (ageing or pregnancy).

Regarding kidney pathologies like diabetes nephropathy, chronic or acute kidney disease, and kidney cancer, the elevation of GDF15 in plasma/urine is beneficial for the kidney function by decreasing renal inflammation and oxidative stress in these pathological situations. In physiological contexts, GDF15 contributes to the homeostatic system in response to a dietary acid load or K^+^ restriction by inducing the proliferation of specific renal cells. All these data clearly establish that GDF15 is a novel factor that has to be taken into consideration when we investigate the pathophysiological mechanisms of kidney diseases or the renal function in physiological situations.

## Figures and Tables

**Figure 1 ijms-25-05956-f001:**
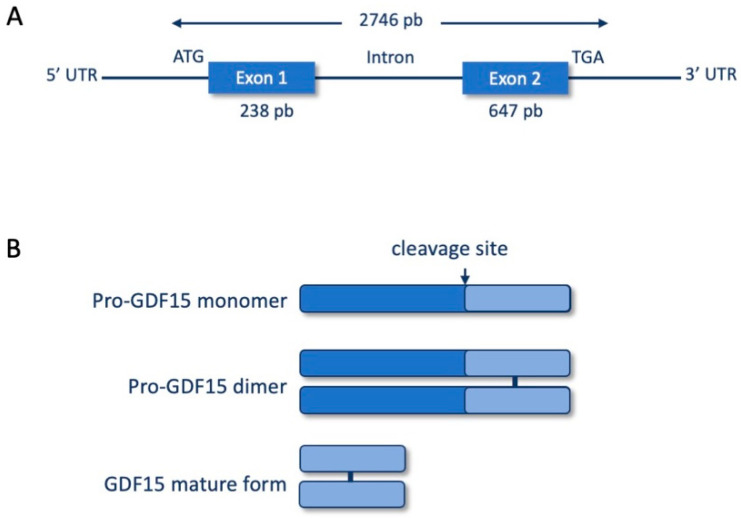
Schematic representation of gene (**A**) and protein (**B**) structures of GDF15.

**Figure 2 ijms-25-05956-f002:**
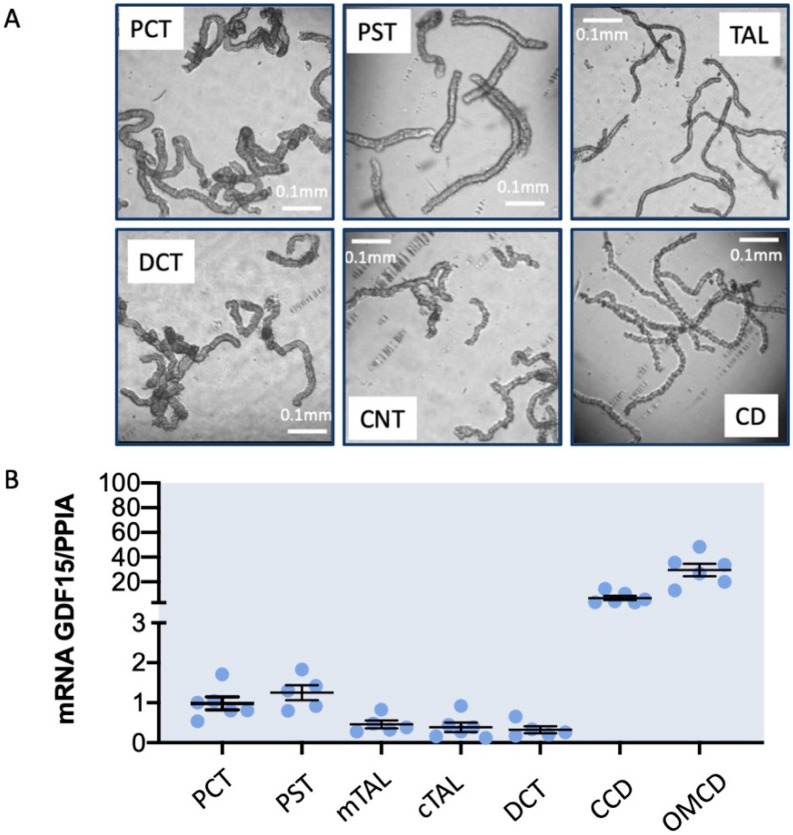
Expression of GDF15 along the nephrons. (**A**) Renal segments isolated under binocular after treatment of kidneys with Liberase (Sigma-Aldrich, St Quentin Fallavier, France) as described in [38] were used for RNA extraction and qPCR analysis as described in [38] for measuring expression of GDF15 (**B**). This expression is normalized with that of a housekeeping gene, cyclophilin (PPIA).

**Figure 3 ijms-25-05956-f003:**
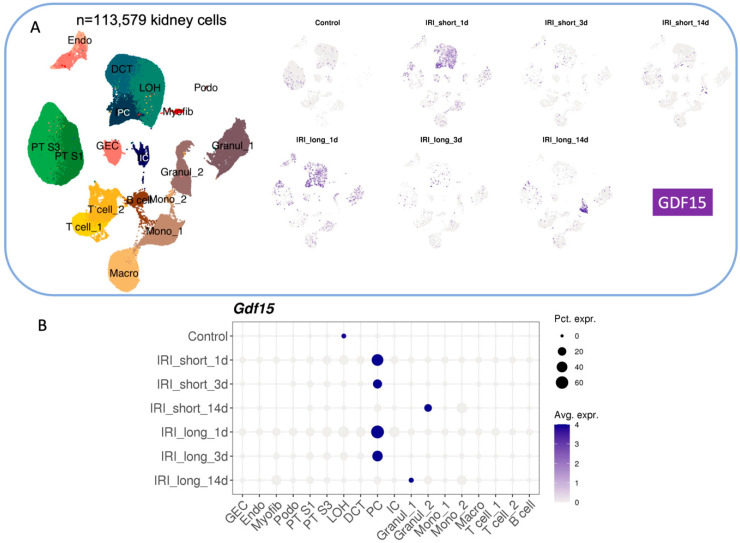
GDF15 in response to ischemia/reperfusion. (**A**,**B**) Analysis by scRNAseq of GDF15 expression in kidney cells after ischemia/reperfusion from http://www.susztaklab.com/Mouse_IRI_scRNA/Genemap.php accessed on 1 April 2024.

**Figure 4 ijms-25-05956-f004:**
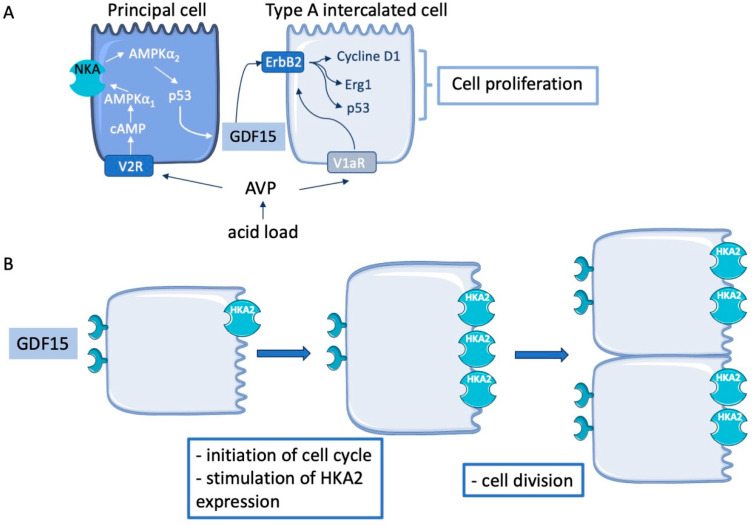
(**A**) Schematic representation of GDF15 production by principal cells and its proliferative effect on type A intercalated cells in the context of acid load. (**B**) Schematic representation of GDF15 effect on type A intercalated cells in the context of K^+^ restriction.

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
