# Peer review of "GDF15, an Emerging Player in Renal Physiology and Pathophysiology"

_ijms, 2024, doi:10.3390/ijms25115956_

Round 1

Reviewer 1 Report

Comments and Suggestions for Authors

This review paper on GDF15 and its regulation function on kidneys is clearly written and comprehensive and represents good systematization of knowledge on subject molecule. 

I suggest to authors to include GDF15 signaling via SMAD Protein Activation in section that considers molecular mechanisms and signaling pathways (please refer to review by Rochette et al., https://doi.org/10.1016/j.tem.2020.10.004

 Also, since this review deals with renal function, it would be quite useful to add a section on influence of disease such as diabetes to kidney, namely diabetic nephropathy and inflammation mediated by AGE/RAGE inflammatory pathway and influence of GDF15 to this kind of disorder (consider Chen et al., 2022, 10.1016/j.lfs.2022.121142

I suggest adding references of newer date to this paper, since they are available.

Author Response

Dear Editors and Reviewers

We thank the reviewers for their comments and corrections that help to improve our manuscript. We hope to have responded satisfactorily to their request.

Reviewer 1: Comments and Suggestions for Authors

This review paper on GDF15 and its regulation function on kidneys is clearly written and comprehensive and represents good systematization of knowledge on subject molecule. 

I suggest to authors to include GDF15 signaling via SMAD Protein Activation in section that considers molecular mechanisms and signaling pathways (please refer to review by Rochette et al., https://doi.org/10.1016/j.tem.2020.10.004

In this review, Rochette et al mentioned that the activation of SMAD pathway by GDF15 is doubtful because of contamination of GDF15 by TGF beta. This is correlated by another review about GFRAL/GDF15 signaling pathway (Tsai et al. Cell Metab 2018).  It perfectly correlates with our statement “RET phosphorylation activates the AKT, ERK1/2, and PKC signaling pathways but not the SMAD signaling pathway, unlike other growth factors in the TGFβ family (for review see (14))”. As done by Rochette et al. and Tsai et al. we now mention the fact that some studies suggest activation of SMAD with the caveat of a possible contamination.

 Also, since this review deals with renal function, it would be quite useful to add a section on influence of disease such as diabetes to kidney, namely diabetic nephropathy and inflammation mediated by AGE/RAGE inflammatory pathway and influence of GDF15 to this kind of disorder (consider Chen et al., 2022, 10.1016/j.lfs.2022.121142

We thank the reviewer for this suggestion and we have included a comment about the Life Science paper by Chen et al. in the “GDF15 in diabetic nephropathy” part.

I suggest adding references of newer date to this paper, since they are available.

We have added few more recent references and we have now 36 references over 98 (almost 40%) that are from the last 5 years. 

Reviewer 2 Report

Comments and Suggestions for Authors

The authors summarize recent findings about the GDF15-role in renal function respectively renal diseases.

The Reviewer has some major comments:

1.     The title does not reflect what the text is about. Maybe instead of renal functions, the authors should aim for ”…renal physiology and pathophysiology”, as more general term.

2.     Figure 2: It remains unclear what kind of structures were shown here and what is the reason for showing these images? Also in combination with the text above, there is not a real context for these images given. Additionally, the abbreviations of the renal segments need to be explained in the figure legend. In the Text part “1.3/GDF15 and kidney”, there is mentioned that the protein is expressed along the tubular segments, but why the picture then? It is 1mm scale bar and thus not possible to display any single proteins or else relevant information here.

3.     In the mentioned text part, the first signaling pathway is mentioned (MAPK pathway). However, this is somehow not fitting into the context here and signaling pathways continue in chapter “GDF15 in chronic kidney disease (CKD) and acute kidney disease (AKI)” on page 5. There were also some pathways described (e.g. PI3K/AKT) but again, the structure of the single chapter appears somehow mixed up. Concrete: It remains unclear whether GDF15 exhibits positive or negative effects during AKI or CKD. The author would strongly recommend to design a table as overview instead of the text format here. For example, using some arrows as indicators for of up or downregulation of all the pathways which are mentioned through the whole text in one table.

4.     Additionally, the Reviewer strongly recommends to focus on IRI in a single chapter, rather than mixing all the pathways in a CKD and AKI chapter. Then figure 3 makes sense and the authors can go through early and late effects of IRI, as obviously the expression of GDF15 is the strongest in early phase.

5.     The conclusion starts with 2 very general sentences, before pointing out the kidney related topics, and then clarifies that most of the studies focusing kidney are observational and descriptive. Furthermore, this conclusion points out that the GDF15 factor is excreted by kidneys and has effects on many other organs, leading to open questions about the GDF15 role during pregnancy and ends up with a very condensed point of a role in K+ retention in one very special pathway. Summarizing, this conclusion has not a connection to what the title promises and let assume, that the whole review is very less kidney related, what is not true in fact. In the Reviewers eyes, the conclusion weakens the whole manuscript, and the whole manuscript itself needs a changed structure, towards “beneficial” and “non beneficial” effects of GDF15 on the kidney, and the same for on “other organs” for example. In the actual way. It is hard to follow and extract what the review should really tell the reader.

Comments on the Quality of English Language

English is ok

Author Response

Dear Editors and Reviewers

We thank the reviewers for their comments and corrections that help to improve our manuscript. We hope to have responded satisfactorily to their request.

The authors summarize recent findings about the GDF15-role in renal function respectively renal diseases.

The Reviewer has some major comments:

  1. The title does not reflect what the text is about. Maybe instead of renal functions, the authors should aim for ”…renal physiology and pathophysiology”, as more general term.

We modified the title accordingly

  1. Figure 2: It remains unclear what kind of structures were shown here and what is the reason for showing these images? Also in combination with the text above, there is not a real context for these images given. Additionally, the abbreviations of the renal segments need to be explained in the figure legend. In the Text part “1.3/GDF15 and kidney”, there is mentioned that the protein is expressed along the tubular segments, but why the picture then? It is 1mm scale bar and thus not possible to display any single proteins or else relevant information here.

We better introduce this Figure now. We think it was important in the context of this review and in a journal of general audience to show how the nephron segments look like instead of drawing yet another scheme of a nephron. The Figure 2 legend, clearly stipulate that the data shown are results of qPCR and the annotation of the y axis of the figure mentioned mRNA GDF15/PPIA. We did not mention “protein”. There are therefore the results of PCR from RNA extracted from isolated tubular segments (the one shown in Figure 2A). We now complete this part by adding a comment and a reference (Liu et al. 2020) on protein expression along the nephron.

  1. In the mentioned text part, the first signaling pathway is mentioned (MAPK pathway). However, this is somehow not fitting into the context here and signaling pathways continue in chapter “GDF15 in chronic kidney disease (CKD) and acute kidney disease (AKI)” on page 5.

The I.1 part is simply organised with 2 subparts, first we described the classical signalling pathways related to GDF15 activation of GFRAL and in a second subpart, we described the ones triggered by GDF15 activation of ErbB2.

The MAP kinase pathway is relevant in the context of this review and needs to be introduced since it is triggered by GDF15 in response to acidosis (see II.1).

There were also some pathways described (e.g. PI3K/AKT) but again, the structure of the single chapter appears somehow mixed up.

The induction of PI3K/AKT pathway was already mentioned in the “I.1/ Signaling pathways related to GDF15” part where we stressed out the fact that GDF15 activation of ErbB2 was shown to induce these pathways. Therefore, the description of GDF15 triggering the PI3K/AKT does not arrive unexpectedly at the page 5 as mention by the reviewer.

Concrete: It remains unclear whether GDF15 exhibits positive or negative effects during AKI or CKD.

It is difficult to follow the reviewer’s assumption here since, in this short paragraph, we have written: “GDF15 deficiency exacerbates renal fibrosis, whereas GDF15 overexpression attenuates fibrotic changes, suggesting a protective role for GDF15 against renal fibrosis”, “…that GDF15 administration attenuates the renal effect of IRI by stimulating expression of Klotho, a nephroprotective factor.” and ”… leading to the suppression of pro-inflammatory cytokines and apoptotic mediators”.

All these sentences clearly indicate that GDF15 has a rather positive effect during CKD and AKI.

The author would strongly recommend to design a table as overview instead of the text format here. For example, using some arrows as indicators for of up or downregulation of all the pathways which are mentioned through the whole text in one table.

We do not think it is necessary to add a Table, the paragraph is concise and the different studies go in the same direction: GDF15 is elevated during CKD and AKI and GDF15 protects the kidney by inhibiting NFkB and limiting the inflammation. We have added a sentence at the end of this part to mention this point without ambiguity  

  1. Additionally, the Reviewer strongly recommends to focus on IRI in a single chapter, rather than mixing all the pathways in a CKD and AKI chapter. Then figure 3 makes sense and the authors can go through early and late effects of IRI, as obviously the expression of GDF15 is the strongest in early phase.

We do not agree with the reviewer since there is a strong link between these two pathological entities. We first start with CKD, then we moved to AKI and we discussed the publication by Valino-Rivas et al. that studied both.

  1. The conclusion starts with 2 very general sentences, before pointing out the kidney related topics, and then clarifies that most of the studies focusing kidney are observational and descriptive. Furthermore, this conclusion points out that the GDF15 factor is excreted by kidneys and has effects on many other organs, leading to open questions about the GDF15 role during pregnancy and ends up with a very condensed point of a role in K+ retention in one very special pathway. Summarizing, this conclusion has not a connection to what the title promises and let assume, that the whole review is very less kidney related, what is not true in fact. In the Reviewers eyes, the conclusion weakens the whole manuscript, and the whole manuscript itself needs a changed structure, towards “beneficial” and “non beneficial” effects of GDF15 on the kidney, and the same for on “other organs” for example. In the actual way. It is hard to follow and extract what the review should really tell the reader.

We agree with the reviewer that the conclusion was not straightforward. We have re-written it in a more simple and clear way.